# Factors to Consider for the Correct Use of γH2AX in the Evaluation of DNA Double-Strand Breaks Damage Caused by Ionizing Radiation

**DOI:** 10.3390/cancers14246204

**Published:** 2022-12-15

**Authors:** Davide Valente, Maria Pia Gentileschi, Antonino Guerrisi, Vicente Bruzzaniti, Aldo Morrone, Silvia Soddu, Alessandra Verdina

**Affiliations:** 1Unit of Cellular Networks and Molecular Therapeutic Targets, Department of Research and Advanced Technologies, IRCCS Regina Elena National Cancer Institute, 00144 Rome, Italy; 2Unit of Radiology and Diagnostic Imaging, Department of Clinical and Dermatological Research, IRCCS San Gallicano Dermatological Institute, 00144 Rome, Italy; 3Unit of Medical Physics and Expert Systems, Department of Research and Advanced Technologies, IRCCS Regina Elena National Cancer Institute, 00144 Rome, Italy; 4Scientific Direction, IRCCS San Gallicano Dermatological Institute, 00144 Rome, Italy

**Keywords:** DNA damage, double-strand breaks (DSBs), DNA repair, γH2AX scoring, ionizing radiation, basal level

## Abstract

**Simple Summary:**

The increase of exposure to ionizing radiation (IR) from medical procedures has prompted research into improving methodologies for the detection of DNA double-strand breaks (DSBs). The measurement of γH2AX by immunofluorescence has become the gold standard for this analysis. However, for the correct use of γH2AX as a biomarker for the assessment of IR-induced DNA DSBs, several exogenous and endogenous conditions that can influence γH2AX levels must be taken into consideration. Here, we describe the conditions leading to H2AX phosphorylation, the most widely used methods for its detection, the principal applications, and the related problems of γH2AX scoring, with particular regard to clinical studies.

**Abstract:**

People exposed to ionizing radiation (IR) both for diagnostic and therapeutic purposes is constantly increasing. Since the use of IR involves a risk of harmful effects, such as the DNA DSB induction, an accurate determination of this induced DNA damage and a correct evaluation of the risk–benefit ratio in the clinical field are of key relevance. γH2AX (the phosphorylated form of the histone variant H2AX) is a very early marker of DSBs that can be induced both in physiological conditions, such as in the absence of specific external agents, and by external factors such as smoking, heat, background environmental radiation, and drugs. All these internal and external conditions result in a basal level of γH2AX which must be considered for the correct assessment of the DSBs after IR exposure. In this review we analyze the most common conditions that induce H2AX phosphorylation, including specific exogenous stimuli, cellular states, basic environmental factors, and lifestyles. Moreover, we discuss the most widely used methods for γH2AX determination and describe the principal applications of γH2AX scoring, paying particular attention to clinical studies. This knowledge will help us optimize the use of available methods in order to discern the specific γH2AX following IR-induced DSBs from the basal level of γH2AX in the cells.

## 1. Introduction

People are continuously exposed to radiation, including ionizing radiation (IR), coming from natural or artificial sources. IRs are the most important and studied radiations for their dangerous effects on biological matter. They induce DSBs, the most serious of DNA damage which, if not correctly repaired by the DNA-damage response (DDR) molecular pathways, can induce genomic instability, chromosome aberrations, and mutations, possibly leading to cancer onset [1,2]. IRs have widespread applications in medicine and their use in clinical settings is continuously increasing, especially in diagnostic instrumental analysis, including X-ray imaging, computed tomography (CT), positron-emission tomography (PET), nuclear medicine, and interventional procedures [3,4]. Only diagnostic X-ray procedures, such as screening tests for cancer, are responsible for about 14% of the total annual exposure to IR worldwide [5,6,7]. According to Report No.160 by the National Council on Radiation Protection and Measurements (NCRP) (https://ncrponline.org/publications/reports/ncrp-report-160-2/ 1 June 2015), in the United States during 2006, the IR exposition from medical procedures constituted nearly half of the total IR exposure and Americans were exposed to IR levels from medicine more than seven times higher compared to the early 1980s. This increased IR exposure has mostly been due to the greater utilization of CT and nuclear medicine in clinical applications. Indeed, the number of CT scans and nuclear medicine procedures performed in the United States during 2006 was estimated to be 67 million and 18 million, respectively. These two procedures alone contributed to 36 percent of the total IR exposure and 75 percent of the medical IR exposure of the U.S. population (https://ncrponline.org/publications/reports/ncrp-report-160-2/ 1 June 2015).

The constant increase of using IRs in diagnostic/radiological tests and nuclear medicine inevitably leads to a potential increase in the risk of DSBs, to which patients are subjected despite the low doses of radiation used in individual diagnostic procedures. Particular attention must be paid to cancer patients, who undergo many radiological investigations repeated over time for follow-up and throughout the treatment course at increasingly shorter time intervals.

Epidemiological studies have shown that exposure to IRs used in radiological diagnostics can increase the risk of stochastic effects [8,9] and that the effective cumulative dose received during the various diagnostic procedures can lead to an increased risk of cancer in somatic cells and of hereditary alterations in germ cells even after months or years [10,11]. It is therefore essential to have sensitive, specific, and noninvasive methods for evaluating the DNA DSBs induced by clinical radiations in exposed individuals in order to correctly assess the risk–benefit ratio and optimizing procedures. However, in order to have realistic evidence of IR-induced DNA DSBs, additional exogenous and endogenous conditions, such as cellular replication, oxidative stress, apoptosis, hypoxia, genome rearrangements, etc., that can induce DNA DSBs and H2AX phosphorylation in physiological and/or pathological conditions, must be taken into consideration.

Several methods have been developed for DNA DSB detection, including the comet assay, the dicentric chromosome aberration test, and the micronuclei method, but none of them are sensitive and specific enough for DSBs nor can they be used on large-scale determinations [12]. The identification of specific markers has allowed for the identification of a sensitive, specific, noninvasive, and large-scale detection of DNA DSBs, which is the measurement of γH2AX (the phosphorylated form of the histone variant H2AX) and its colocalization with the P53-binding protein1 (53BP1) by immunofluorescence, which has become the gold standard for the detection of DSBs [13,14,15]. 

In this review, the role of γH2AX in DNA DSB repair and the different conditions responsible for its induction will be summarized as well as the most widely used methods for its detection will be discussed. An overview of the applications of γH2AX scoring with particular regard to clinical studies will also be described.

## 2. Basic Mechanism of H2AX Phosphorylation

The H2A histone family member X (H2AX) is a variant of the histone protein H2A. It is involved in nucleosome formation and chromatin remodeling and is an essential protein for DNA DSB repair [16]. Human cells are constantly exposed to different DNA-damaging factors, both exogenous and endogenous, that induce DSBs (Figure 1) and respond to the onset of damage by H2AX phosphorylation and the activation of the repair pathway [17]. Exogenous factors are represented by environmental and clinical factors including background radiation, radiation accidents, occupational exposure, chemical mutagens, heat, medical IR exposure, and chemotherapeutic drugs [18,19,20,21,22,23,24,25]. Endogenous factors include genome rearrangements such as V(D)J recombination or meiotic recombination, free radicals produced during oxidative stress, telomere shortening in senescent cells, DNA fragmentation during apoptosis, and hypoxia [26,27,28,29,30,31,32,33,34,35]. 

DNA DSB repair is a very important biological mechanism that recognizes and corrects damage or abnormalities on the genome and where H2AX phosphorylation has a key role in the repair process. The two major pathways for DNA DSB repair are homologous recombination (HR) and nonhomologous end-joining (NHEJ). The first leads to accurate repair, while the second can be mutagenic; both involve γH2AX [36,37]. 

Since its discovery, γH2AX has been considered and utilized as a marker of DSBs and different methods have successfully been developed for its detection, including Western blotting, ELISA, flow cytometry, and immunofluorescence [13]. This has allowed to evaluate the risk for individuals exposed to IR [38] and predict individual response in the clinical setting [39].

H2AX is a critical sensor which provides a recruitment site for the mediators involved in DSB repair. One of the key steps in signaling and initiating the repair of DSBs is H2AX phosphorylation at Ser139, which involves 2–30 Mbp around the break, 10^4^ to 10^5^ nucleosomes, and 10^2^ to 10^3^ H2AX proteins, providing binding sites for the additional DDR components [40,41,42,43]. H2AX-Ser139 phosphorylation is catalyzed by the Ataxia-telangiectasia-mutated (ATM) protein kinase, which is activated in response to DSBs by autophosphorylation at Ser1981. The DNA-dependent protein kinase (DNA-PK) and Ataxia-telangiectasia and Rad3-related (ATR) protein kinase, belonging to the phosphatidylinositol-3-OH-kinase-like family of protein kinases (PIKKs), also phosphorylate H2AX [44,45,46]. After the initial phosphorylation of H2AX at sites flanking DSBs, numerous repair proteins are recruited, including the Mre11–Rad50–Nbs1 (MRN) protein complex, 53BP1, and cohesins, which maintain the DNA ends in close proximity during repair [47,48,49,50] (Figure 2). This leads to a further activation of ATM and the phosphorylation of H2AX, forming γH2AX foci. 

For its early involvement, γH2AX is considered a marker of very-early response to DSBs. Immunofluorescence analysis showed that γH2AX foci are detectable at the sites of DSBs within 3 min after DNA damage stimulation and continue to increase until a plateau is achieved in 10–30 min, where the number of γH2AX foci has been shown to correlate with the number of DSBs [42]. Then, a kinetic disappearance is observed up to about 24 h, where γH2AX levels return approximatively to baseline. A typical example of γH2AX foci is shown in Figure 3, where human primary fibroblast cells (HFs) irradiated with 0.2 Gy, fixed at different times (0, 0.5 h, 1 h, 3 h, 24 h), and stained for γH2AX with a specific antibody have been analyzed by immunofluorescence and manual counting. The figure shows γH2AX detection within 30 min after irradiation and a kinetic disappearance up to about 24 h, where γH2AX levels return approximatively to baseline (AV, unpublished data).

HFs were grown in Dulbecco’s modified Eagle’s medium (DMEM) supplemented with 10% fetal calf serum and antibiotics (1% penicillin–streptomycin) in a humidified 5% CO_2_ atmosphere at 37 °C. A total of 80,000 cells were seeded onto cover glasses in sterile noncoated six-well plates, grown until the day of treatment (70% confluence) and irradiated with 0.2 Gy (Irradiator IBL 437C, Cis bio International). Then, the cells were fixed at different times (0, 0.5 h, 1 h, 3 h, 24 h) in 3.7% paraformaldehyde for 10 min and permeabilized using 0.25% Triton X-100 for 10 min, followed by blocking with 5% BSA solution for 1 h. After, they were stained for γH2AX with specific mouse monoclonal antibody against phosphohistone H2AX (Ser139) and analyzed by immunofluorescence. 

This trend depends on the individual repair capacity which affects the kinetic of DSB repair and the disappearance of γH2AX foci [51]. It has been proposed that after repair, γH2AX is removed by dephosphorylation by the following protein phosphatases: Protein phosphatase 2 A (PP2A), Protein phosphatase 4 C (PP4C), Protein phosphatase 6 (PP6), and wild-type p53-induced phosphatase 1 (WIP1) [52,53,54,55] (Figure 2). However, the redistribution of γH2AX in chromatin by acetyltransferase-mediated histone exchange, replacing γH2AX with Histone H2A.Z (H2AZ), has also been proposed [56]. In addition to γH2AX, 53BP1 is another key DSB-responsive protein promoting the repair of DSBs by NHEJ while preventing HR. Thus, 53BP1 is recruited by γH2AX and in turn acts as a recruiter for other DDR proteins [57,58], orchestrating the choice of the DSB repair pathway. 

While the induction of DSBs always causes the phosphorylation of the histone H2AX, the presence of γH2AX foci should not always be considered a marker of DSBs. Physiological conditions, including cell replication, oxidative stress, apoptosis, hypoxia, genome rearrangements such as V(D)J recombination, and spermatogenesis, are characterized by H2AX phosphorylation which is not always matched with DSBs [26,27,28,29,32,33,59,60]. 

Based on the morphology, different γH2AX foci populations have been identified and different functional roles have been attributed. Two populations of γH2AX foci distinguished by the size and colocalization with damage-repair proteins, including 53BP1, were first described in 2005 in untreated, proliferating mammalian cells [61]. As shown in Figure 4 (AV, unpublished data), there is a prevalent population of small foci that do not colocalize with 53BP1, and a small population of large foci that do colocalize with 53BP1. The large foci appear similar to the foci induced by DSBs following IR exposure and represent naturally occurring DNA DSBs.

The cells were cultured, irradiated, and fixed after 1 h from irradiation, as described in Figure 3. Immunofluorescence was performed using specific mouse monoclonal antibody against phosphohistone H2AX (Ser139) and rabbit polyclonal antibody against 53BP1 and analyzed by immunofluorescence. 

Since IR-induced DSBs and γH2AX-foci formation can overlap with a background of pre-existing γH2AX foci which accumulate in response to different exogenous and/or endogenous conditions, their existence must be recognized and considered for the correct biodosimetry of IR-induced DSBs, as summarized in the following paragraphs.

## 3. Conditions That Affect the Observed Level of H2AX Phosphorylation

### 3.1. H2AX Phosphorylation and the Cell Cycle

In 2016, a study assessing the association between γH2AX, 53BP1, and DNA replication in the different phases of the cell cycle demonstrated that low levels of γH2AX are not induced by DSBs [62]. Based on size and brightness parameters, Rybak et al. [62] discriminate between small and low-brightness foci (“dim” foci), containing fewer γH2AX molecules, and large and high-brightness foci (“bright” foci), the latter related to the presence of DSBs. The analysis of the “dim” and “bright” γH2AX foci in untreated cells and following exposure to DNA topoisomerase inhibitors inducing DNA damage showed that the nature of these foci is different based on the relationship with replication factors and 53BP1 foci. In particular, in untreated S-phase cells, where DNA damage is mainly induced by endogenous oxidants, the “bright” γH2AX foci weakly correlate with the number of DNA replication sites (i.e., DNA sites on which the replication machinery is active and 5-Ethynyl-2′-deoxyuridine (Edu) is incorporated) and do not occur preferentially near replicating DNA. They are closely associated with 53BP1, suggesting a correlation with the sites of the DSBs. On the contrary, “dim” γH2AX foci are more numerous and correlate with the number of DNA replication sites, although they do not occur preferentially near replicating DNA and, above all, are not associated with 53BP1. This suggests that “dim” foci do not represent a DNA-repair response. 

In cells treated with topoisomerase inhibitors, the number of “bright” γH2AX foci increases and correlates with the number of DNA replication sites. Most of these foci correlate with 53BP1, indicating a link with DSBs. Conversely, “dim” γH2AX foci are not associated with 53BP1 and are not found near the replication sites, thus indicating that low γH2AX is also unrelated to DSBs in treated cells. Therefore, the available evidence suggests that these γH2AX signals are induced by mechanism(s) other than the formation of DSBs. 

In the past few years, the cell-cycle-dependent expression of γH2AX has been described. MacPhail and colleagues demonstrated an increase of γH2AX levels during the progression from the G1 to S phase in cells exposed to DNA DSB-inducing agents [63]. McManus and Hendzel [61] showed that, in untreated mammalian cells, H2AX phosphorylation increases depending on the phase of the cell cycle, with a maximal ATM-dependent H2AX phosphorylation during mitosis. In particular, they observed an increase of γH2AX when cells entered the early S phase, and which kept increasing when the cells progressed into the mid- and late-S-phase. Moreover, when cells entered in the G_2_/M phase, the signal of H2AX phosphorylation kept increasing further. During mitosis, maximal levels were reached at the metaphase. In the anaphase, the γH2AX signal decreased until it reached a steady state after cytokinesis. Cell-cycle dependence of γH2AX was also shown by Tanaka and colleagues [28] in peripheral blood mononuclear cells (PBMCs), both in G_0_ and during mitogenic stimulation. Their results demonstrated that G_0_ PBMCs characterized by a minimal rate of oxidative metabolism exhibit a very low H2AX-phosphorylation level, slightly above the nonspecific level. Phytohaemagglutinin (PHA) treatment induced an increase in the transcriptional and translational activity, mitochondria number, and oxidative metabolism, as well as an increase of the DNA replication [64,65,66]. The authors observed a high increase in H2AX phosphorylation together with the cellular DNA content in treated cells. Specifically, γH2AX immunofluorescence after 72 h treatment was 10-fold higher (145.0 vs. 15.5) in stimulated G_1_ cells compared to unstimulated G_0_ cells. The difference was greater for the S and G_2_M cells, whose mean immunofluorescence level of γH2AX was about 17-fold (263.4 vs. 15.5) and 27-fold (418.8 vs. 15.5) higher, respectively, compared to unstimulated G_0_ cells. In addition, ATM, which is usually not activated in G_0_ PBMCs, was markedly activated upon mitogenic stimulation. Since these increases correspond to a rise of endogenously generated reactive oxygen species (ROS), and γH2AX and activated ATM are strongly associated with cellular metabolism and oxidant rates in PBMCs undergoing mitogenic stimulation, the authors provided evidence that activated ATM and γH2AX reflect the oxidative DNA damage induced by the reactive oxygen species during progression through the cell cycle following PHA stimulation [67,68]. 

Hernandez and colleagues [69] quantified γH2AX foci in combination with analysis of the cell-cycle phases in human mammary epithelial cells and showed that γH2AX foci scoring and labeling patterns are related to the cell-cycle phases. Using a double immunodetection of bromodeoxyuridine (BrdU) and γH2AX, an irregular γH2AX nuclear staining across the nucleus that does not produce distinct foci, was observed in BrdU-positive cells (S-phase nucleus), whereas discrete γH2AX foci were observed in cells negative for BrdU. The γH2AX staining pattern was different in M-phase cells: staining was also pan-nuclear, but brighter and more homogeneous. All these observations demonstrated an involvement of γH2AX in the surveillance of DNA replication and contribute to explaining the background of the γH2AX signal in the cells. 

### 3.2. H2AX Phosphorylation and Oxidative Stress

Oxidative stress is the consequence of an imbalance between the production and accumulation of ROS in cells and tissues. It is a harmful process that can negatively affect several cellular structures, including DNA [70,71,72,73,74,75]. The width of DSBs by endogenous oxidants varies widely [76,77,78,79,80]. To provide a better understanding, just consider that about 50 DSBs per nucleus are generated during a single cell cycle in human cells in the absence of external stimuli [80]. Consequently, despite the intervention of the DNA-damage repair systems, a background level of H2AX phosphorylation can be observed in cells in the absence of any exogenous genotoxic stimulus. It reflects the cell response to oxidative DNA damage induced by endogenous oxidants produced by metabolism during the cell-cycle progression. It has been shown that, in untreated normal and tumor cells, a fraction of the histone H2AX remain phosphorylated and the extent of this constitutive H2AX phosphorylation depends on cell type and cell-cycle phase [32,63,67,81]. While in interphase H2AX, phosphorylation reflects a DNA-damage response involving the presence of DSBs, these events in mitotic cells may also be associated with torsional stress following chromatin condensation that may represent the signal triggering H2AX phosphorylation even in the absence of DSBs [61,81,82]. 

### 3.3. H2AX Phosphorylation and Apoptosis

H2AX phosphorylation occurs following DNA apoptotic fragmentation as a result of DSBs [60]. Lu and colleagues [59] reported that H2AX may contribute to the apoptotic process in ultraviolet (UV)-stimulated cells. Specifically, they demonstrated that UVA irradiation strongly induces H2AX phosphorylation mediated by c-Jun N-terminal kinase (JNK) and the association with apoptosis. They have been the first to show that JNK-mediated H2AX phosphorylation is required for the apoptosis occurring through the caspase-3/caspase-activated DNAse (CAD) pathway. Solier and collaborators induced apoptosis by the TNF-related apoptosis-inducing ligand (TRAIL) and analyzed single cells by confocal immunofluorescence microscopy [83,84,85]. They observed three staining patterns for γH2AX during this type of apoptosis: initially, a ring staining without massive alteration of nuclear size, which is detected in the early apoptotic cells; then, a pan-staining of the nucleus, which conserves its overall morphology and size; finally, a pan-staining that persists within apoptotic bodies. The ring staining represents an epigenetic landmark of early apoptosis and differs from the typical focal pattern of DSB foci produced by IR. It is not specific for TRAIL-induced apoptosis, but it is also induced by anticancer agents [84,86,87,88,89,90]. Moreover, the authors observed an apoptotic ring in both tumor and primary cells, demonstrating that it is a ubiquitous process.

### 3.4. H2AX Phosphorylation and Hypoxia

Hypoxia is a common feature of solid tumors resulting from inadequate oxygen delivery to the tissues, either due to low blood supply (inadequate vascularization) or low oxygen content in the blood. Severe hypoxia can induce a DNA-damage-like response, involving ATR and ATM activation and subsequently the H2AX phosphorylation [34,35].

In human cancers, the transient interruption of blood flux can alternate with its rapid restoration, leading to a transient hypoxia [91,92]. Giaccia’s group [34] have shown that histone H2AX phosphorylation resulting from hypoxia is ATR-dependent and is maintained in response to reoxygenation-induced DNA damage in an ATM-dependent manner. Later, Wrann et al. [93] demonstrated that hypoxic-inducible factor (HIF)-1 and HIF-2 are involved in the phosphorylation of H2AX under chronic hypoxic conditions, i.e., 0.2% O_2_. They analyzed hypoxic γH2AX induction in a range of cancer cell lines and demonstrated that H2AX phosphorylation is delayed in HIF-1α-deficient mouse embryo fibroblasts (MEFs) and after HIF-1α or HIF-2α knockdown in HEK293 cells, where there is a further decrease when both HIFs are knocked down. Conversely, in 786-O cells constitutively expressing HIF-2α, H2AX phosphorylation increases. 

### 3.5. Environmental Factors and Lifestyle Inducing H2AX Phosphorylation

As described above, many common cellular processes may damage DNA and result in DSBs and γH2AX foci formation, even in the absence of DNA damaging specific external agents [29,61,63,82,94]. Moreover, individuals can commonly be exposed to environmental factors such as heat and background environmental radiation that induce DSBs and H2AX phosphorylation [18,19,25]. In addition, individual factors and lifestyle can influence γH2AX levels [95,96,97] (Figure 1). Sharma and colleagues [98] analyzed the effect of age, gender, race, ethnicity, and alcohol use on the endogenous H2AX phosphorylation in a population of 94 healthy volunteers. They observed that race, ethnicity, and alcohol use significantly affect γH2AX endogenous levels and showed that the marker increases with age. They also observed that endogenous levels positively correlate with IR-induced γH2AX response at 30 min and negatively with residual γH2AX foci at 24 h, demonstrating that basal endogenous levels of γH2AX also affect the kinetics of DNA DSB repair. 

The association between smoking and γH2AX has been shown in several studies using in vitro cellular models [99,100,101]. Ishida and colleagues [102] analyzed PBMCs from twenty-seven young healthy volunteers (12 male nonsmokers, 15 male active smokers) and demonstrated that the number of γH2AX foci per cell is significantly higher in smokers than in nonsmokers (*p* < 0.0001). In addition, the percentage of γH2AX foci-positive cells was significantly higher in smokers than in nonsmokers (*p* = 0.0005). They also demonstrated that smoking cessation results in the reversal of formed DSBs and H2AX phosphorylation to levels comparable to those seen in nonsmokers. Eventually, they observed a strong correlation between the amount of γH2AX foci and exhaled CO levels. 

All these factors cause a basal level of γH2AX signal that shows a high interindividual variation [13]. Ismail and colleagues [103] analyzed the γH2AX signal after the in vitro treatment of blood samples from ten individuals by enediyne calicheamicin γ1 (CLM), an antitumor drug causing strand scissions. They observed a nearly 2-fold variation in the γH2AX signal. To confirm this interindividual variation, they irradiated the blood from twenty patients with 8 Gy and analyzed γH2AX levels in PBMCs. Moreover, under these conditions, they observed a marked variation in the γH2AX signal among individuals. The authors speculated that this may be due to differences among individuals in the efficiency to convert DSBs in γH2AX foci or in the number of γH2AX molecules produced per DSB, although they do not exclude it may be, at least in part, linked to IR sensitivity.

## 4. Methods for DSB Assessment and Current Problems

As described above, γH2AX is a specific molecular marker for DSBs and DNA-damage response and is microscopically visible as discrete foci after specific antibody labeling by immunological techniques [42,104,105,106]. Therefore, the scoring of γH2AX foci has become a widely used measure for evaluating DSBs and DNA repair [69]. 

Quantification of γH2AX foci can be performed either manually, by counting nuclear foci detected by confocal or epifluorescence microscopy and shown on images after capturing them, or automatically, by evaluating the total γH2AX intensity by flow cytometry [69]. 

Microscopic observation and manual quantification represent the most widely used method thanks to its sensitivity and specificity, even if it is time-consuming and subject to the interpretation of the investigator. Flow cytometry represents a rapid method able to discern cell-cycle-dependent H2AX phosphorylation, but it is less sensitive [13]. Since the manual scoring of γH2AX takes too long in large-scale studies, several software programs have been developed which permit foci counting, as well as focus size definition [107,108]. Open-source programs, such as FociCounter and CellProfiler, are used to analyze images after their capture [109,110]. CellProfiler is faster than FociCounter because it does not require treating each image and cell nuclei individually and provides more information about the γH2AX foci and the cell nuclei. In response to the recognized need for methods suitable for large-scale analyses, a logical approach is the complete automation of standard biodosimetric assays that are currently performed manually. The rapid automated biodosimetry tool (RABiT) developed at the Center for High-Throughput Minimally Invasive Radiation Biodosimetry (CHTMIRB) is designed as a completely automated robotically based system to score γH2AX fluorescence in PBMCs derived from a single drop of blood. The automation of PBMC isolation, the immunolabeling of γH2AX, and high-speed imaging allows the analysis of up to 30,000 samples per day [98,111,112,113]. 

However, these methods still have some problems to overcome, such as counting “touching nuclei” or excluding qualitative factors, in particular the variations in the γH2AX foci pattern during the cell cycle. Genescà’s group [69] set up a method (the spot-counting system) for a specific determination of γH2AX radio-induced foci by combining their detection with a cell-cycle analysis. They included in the γH2AX immunofluorescence the cell-cycle markers BrdU, pericentrin, and nuclear area measurements in order to assess automatic scoring of radio-induced γH2AX foci in cells sorted by the cell-cycle phase. This system is able to detect small differences in γH2AX foci number, since it can score a high number of nuclei.

Additionally, the technical issues of γH2AX foci quantification methods have to be considered. An important aspect is related to physical and geometrical limitations due to foci intrinsic extension and to the fluorescence microscope limit (i.e., finite resolution, multiple plane foci position, and/or overlap). Moreover, energy deposition by radiation can occur in different ways based on the exposure source, determining the different distribution of DSBs. For example, high-LET radiation generates very close damage sites forming foci that produce a single-focus signal. All these factors may determine the foci signal saturation, producing misquantification and making it not possible to assume a 1:1 correspondence between the foci count and the DSB [114,115,116]. 

To overcome this, computational tools to predict the real extent of the DNA DSB damage have been developed. Taking into account the different parameters that determine foci saturation, these approaches quantify the miscounting of DNA DSBs after the analysis of immunofluorescence images and provide a tool for a correct interpretation of the DNA DSB data [116,117].

## 5. Applications of γH2AX in Environmental Risk Detection and in Clinical Procedures

γH2AX is considered an indicator of environmental risk, and the most obvious application of its measurement is the assessment of DSBs induced by IR exposure. Historically, the first field of application was related to accidental exposures. More recently, it is extended to radiotherapy and radiodiagnosis. Actually, γH2AX allows to determine DSBs due to external radiations from high-background-radiation areas, space travel, medical radiation treatments, occupational exposures, and radiation accidents [18,19,21,22,23,118,119], or internal, as during radioisotope therapy, where radioisotopes are administered via infusion or ingestion [120]. In addition, γH2AX allows to detect genotoxic potential of DNA-damaging compounds such as cigarette smoke, polycyclic aromatic compounds, electromagnetic fields, microwaves from mobile phones, and extreme heat [99,121,122,123,124,125,126]. 

In the clinical field, γH2AX has various applications. It is used as an indicator of cell death induced by chemotherapeutic agents [127], as a marker of DSBs in human lung adenocarcinoma cells exposed to tobacco smoke [99], and as a tool for evaluating the effectiveness of anticancer therapies and also for differential diagnosis. Specifically, Wasco and Pu [128] suggested γH2AX as a useful adjunct in differential diagnoses of metastatic renal cell carcinoma. 

Another interesting application for the use of γH2AX foci is as an in vivo biodosimeter through their measurement in PBMCs in people undergoing diagnostic radiations, particularly computed tomography (CT). This medical diagnostic procedure has, in fact, greater use involving exposure to small, but not negligible, radiation doses. Loöbrich and colleagues [5] analyzed γH2AX foci in patients who underwent CT of the thorax and/or abdomen and showed a linear relationship between the γH2AX foci number, the exposure dose, and the exposed body surface. In one patient who had previously shown symptoms of radiosensitivity, they detected a significantly higher number of foci than in the other subjects. This effect is related to a defect in the DNA-repair mechanisms, suggesting the possible use of γH2AX as a biomarker of susceptibility as well as exposure. Halm and colleagues observed a dose-dependent increase in γH2AX foci also in the PBMCs of young children (3–21 months) who underwent CT examination [129].

γH2AX foci detection has been used also for monitoring DSBs in interventional radiological procedures. Beels and colleagues [130] used γH2AX foci in PBMCs for the assessment of individual DNA radiation damage in pediatric patients undergoing cardiac catheterization. Kuefner et al. [131] also showed an increase of γH2AX foci, ranging between 0.3 and 1.5 foci/cell, in patients who underwent angiographic procedures. Finally, γH2AX is employed to monitor the normal tissue toxicity and predict patient response to radiotherapy [132].

## 6. Conclusions

Radiation biodosimetry exploits γH2AX detection and is applied in many fields. In the last few decades, exposure to IR from medical procedures, including diagnostic radiology and radiotherapy, has increased the interest in this field and has prompted research into improving methodologies for damage detection. Several endogenous conditions described above induce H2AX and many factors influence γH2AX basal levels that coexist and overlap with γH2AX levels induced by exogenous stimuli. This issue must be considered carefully, especially when DNA damage is induced by low doses of radiation, as in the diagnostic setting. 

Several manual and automatic methods have been developed for γH2AX detection. Manual scoring is the most widely used method thanks to its sensitivity and specificity, even though it is time-consuming, subject to the interpretation of the investigator, and not suitable for large-scale studies. However, the manual method allows to distinguish basal and induced γH2AX on the basis of current knowledge. It is quite the reverse for automated methods, while allowing large-scale studies and excluding the qualitative factors responsible for the differences in the background levels of the γH2AX signal. Through the use of manual scoring, an expert investigator can distinguish the different foci based on morphology: small and low-brightness foci not associated with 53BP1 do not represent a response to DSBs, whereas large and very-bright foci associated with 53BP1 represent it. The use of cell-cycle markers, including BrdU and pericentrin or nuclear area measurements in γH2AX immunofluorescence or flow cytometry help the user to discriminate increments of foci due to cell-cycle progression from those that are radio-induced. γH2AX foci induced during apoptosis can be microscopically distinguished from radiation-induced γH2AX foci by a characteristic staining pattern, ring staining.

Currently the immunofluorescence and the use of γH2AX as a biomarker allow to evaluate the DSBs in different situations. The need for an evaluation in the clinical/diagnostic field where the IR uses doses that are low is satisfied by the sensitivity of the method, but, in this case, the unspecific signal due to the particular conditions described in this review is more conditioning and may drive to an overestimation of DSBs. Thus, when low-dose damage is evaluated, it is particularly important to discriminate an unspecific signal from the foci pattern due to IR exposure. In this case, the manual method is more suitable, as it allows a critical evaluation, assuming that the operator knows how to distinguish between an unspecific signal and induced foci.

In this review, we have summarized the conditions that induce H2AX phosphorylation, related to both specific exogenous damaging stimuli and basic environmental factors, lifestyles, and cellular states. This knowledge will lead us to a more critical use of the techniques available for assessing DSBs, which are based on the use of γH2AX as a biomarker.

## Figures and Tables

**Figure 1 cancers-14-06204-f001:**
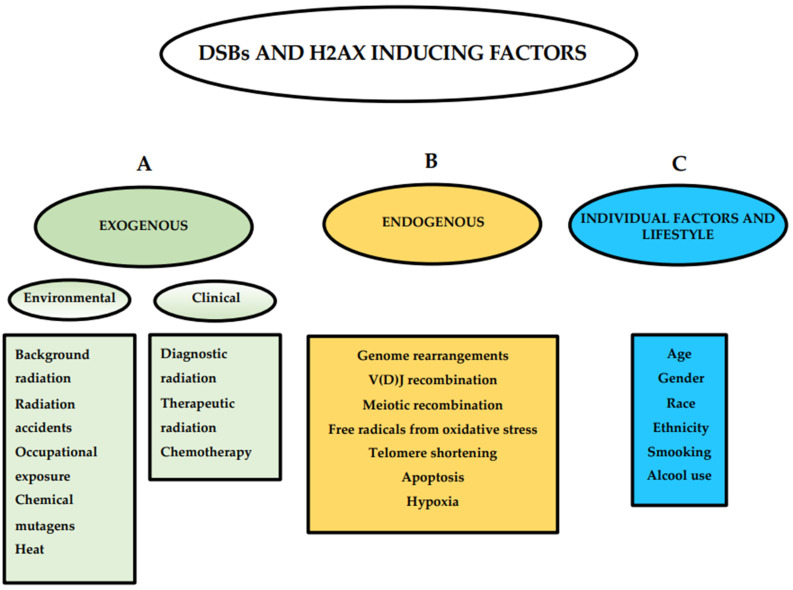
Double-strand break (DSB)- and H2AX-inducing factors. Exogenous, endogenous, individual factors, and lifestyle inducing DSBs and H2AX are described.

**Figure 2 cancers-14-06204-f002:**
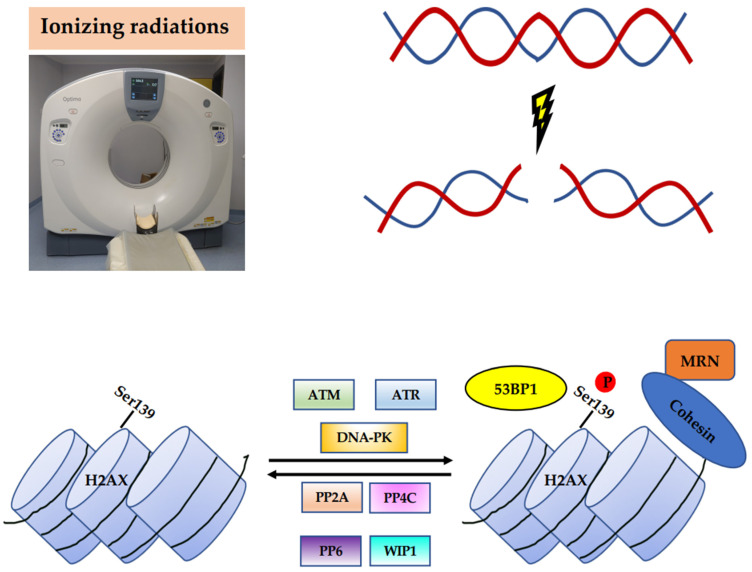
Phosphorylation of H2AX in the DSB response. Upon DSBs induced by IR, H2AX activity is regulated by Ser139 phosphorylation/dephosphorylation cycles. Phosphorylation is catalyzed by ATM, ATR, and DNA-PK. When H2AX is phosphorylated at the sites flanking DSBs, different repair proteins are recruited, including the MRN protein complex, 53BP1, and cohesins. After repair, γH2AX is removed by dephosphorylation catalyzed by the protein phosphatases PP2A, PP4C, PP6, and WIP1.

**Figure 3 cancers-14-06204-f003:**
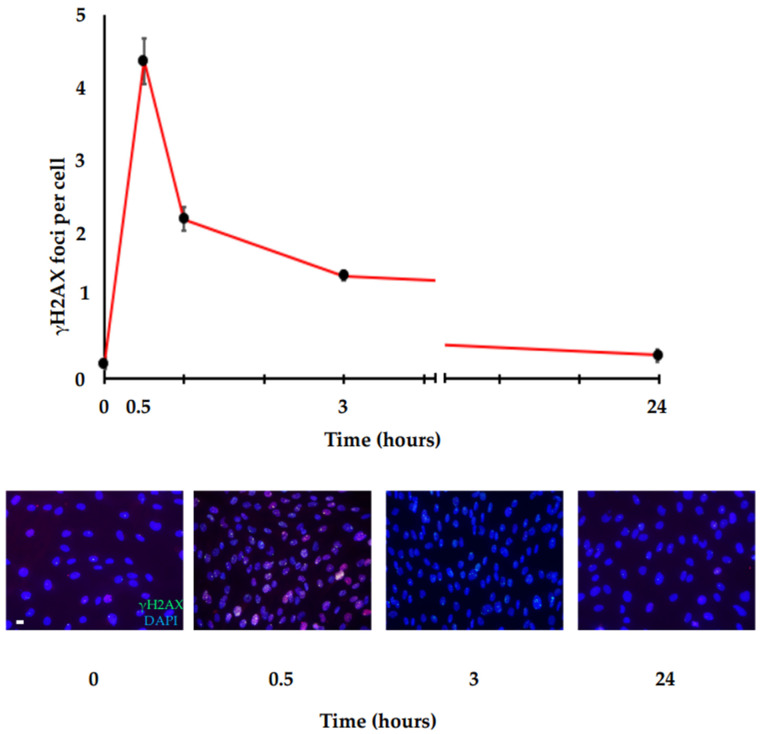
γH2AX foci repair kinetic. Immunofluorescence analysis of γH2AX in HF cells irradiated with 0.2 Gy. Following DNA damage, γH2AX is quickly induced and reaches the maximum peak in a very short time (about 30 min). Then, the γH2AX levels decrease more slowly until they reach the basal levels at about 24 h. Top panel: graph showing the amount of γH2AX foci/cells at the indicated time points after IR stimulus. Bottom panel: representative immunofluorescence images at the indicated time points after IR stimulus (scale bar is 10 µm).

**Figure 4 cancers-14-06204-f004:**
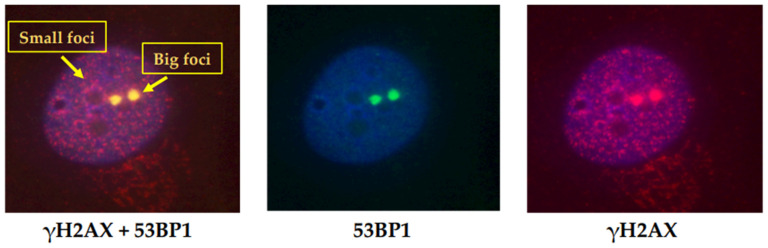
Two different populations of γH2AX foci. Immunofluorescence analysis of γH2AX and 53BP1 in HF cells irradiated with 0.2 Gy. Two populations of γH2AX foci, distinguished by the size and colocalization with DNA DSB repair proteins, are shown. The small foci are more numerous and do not colocalize with 53BP1, while the large foci represent a small population and colocalize with 53BP1.

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
