# Peer review of "Factors to Consider for the Correct Use of γH2AX in the Evaluation of DNA Double-Strand Breaks Damage Caused by Ionizing Radiation"

_cancers, 2022, doi:10.3390/cancers14246204_

Round 1

Reviewer 1 Report

The review Factors to consider for the correct use of the gamma-H2AX in the evaluation of DNA damage caused by ionizing radiation” presents an overview on the use of this kind of DNA damage foci and on the limitations to consider for its use in bio-dosimetry applications.

This reviewer has major concerns about the robustness of this manuscript.

In general, the review can result misleading in several parts of the introduction, as the authors refer to a specific kind of DNA damage, the double-strand breaks, as general “DNA damage”. Despite specified in some paragraphs (e. g. line 115), a review should be clear and introduce the context from the beginning.

Several types of damages are induced by different endogenous and exogenous stress agents (mismatches, single-strand brands, base and nucleotide damages etc…), but repaired by cells without the involvement of gamma-H2AX foci (see lines 86-87).

All the figures related to foci appearance and morphology lack of description to understand what are the conditions from which the foci arose. Information on cell line, radiation type, dose, irradiation setup are missing, making the interpretation of the text and of the images confusing and again misleading.

As the aim of the review is to show how these foci can be used to detect and quantify radiation-induced DSBs, the authors must provide the radio-biological details of the irradiation conditions. The manuscript should be therefore revised to allow a straightforward understanding of the phenotypes related to gamma-H2AX induction and repair.

For example, in Fig.3, which exposure conditions have been used? Which dose? Different radiation modalities could lead to different induction-recovery curves (fractionated therapy, for example).

Another caveat to this review concerns the fact that, while it is important to consider the baseline in gamma-H2AX foci for proper quantification of the radio-induced DNA DSBs (and the factors are quite well reported), no mention is made to the limitations that the described techniques to detect/count foci (IF and flow cytometry) bring along.

Following high-doses of radiation (for example after radiation accidents, exposure to high doses of space radiation, etc…), or exposures with high-LET radiations (radiotherapy), experimental observations using high-resolution microscopy have proved that the common readouts used to quantify DNA DSBs lead to mis-quantifications.

Moreover, foci miscounting due to saturation of related signal has been reported in different papers:

·         Ivashkevich, A et al. Use of the γ-H2AX assay to monitor DNA damage and repair in translational cancer research. Cancer Letters, 2012.

·         Avondoglio, D et al. High throughput evaluation of gamma-H2AX. Radiat Oncol,  2009.

·         Tommasino, F et al. Induction and Processing of the Radiation-Induced Gamma-H2AX Signal and Its Link to the Underlying Pattern of DSB: A Combined Experimental and Modelling Study. PLOS ONE 2015.

The need of tools to help us quantifying DSB induction, and repair, led to the development of computational tools to predict the real extent of the lesions in biological matter, and in particular in the DNA. Several authors propose then computational approaches to quantify the miscounting of DNA DSBs after analysis of immunofluorescence images from microscopy:

·         Tommasino, F et al. Induction and Processing of the Radiation-Induced Gamma-H2AX Signal and Its Link to the Underlying Pattern of DSB: A Combined Experimental and Modelling Study. PLOS ONE 2015.

·         Barbieri, S et al. Predicting DNA damage foci and their experimental readout with 2D microscopy: a unified approach applied to photon and neutron exposures. Sci Rep. 2019.

I think the authors need to include this point to properly assess the pros and cons of using gaamm-H2AX foci in bio-dosimetry. This reviewer believes that the review at its current state is not ready for publication. 

Author Response

We are grateful to reviewer for the constructive critics and suggestion.

1)         The review can result misleading in several parts of the introduction, as the authors refer to a specific kind of DNA damage, the double-strand breaks, as general “DNA damage”. Despite specified in some paragraphs (e.g. line 115), a review should be clear and introduce the context from the beginning.

As suggested by the referee, we specified throughout the text the analysis of DSBs rather than using a more general definition such as “DNA Damage”. We changed also the title in “Factors to consider for the correct use of γH2AX in the evaluation of DNA double strand breaks damage caused by ionizing radiation”.

2)         All the figures related to foci appearance and morphology lack of description to understand what are the conditions from which the foci arose. Information on cell line, radiation type, dose, irradiation setup is missing, making the interpretation of the text and of the images confusing and again misleading.

Though our intent was just to show a typical curve of γH2AX signal dynamics after irradiation without providing specific experimental data, we fully agree with the referee. Thanks to his suggestion we realized that lack of experimental details might be misleading. Information relative of experimental conditions for the data in figure 3 and figure 4 has been added in the Figure legend and in the text.

3)         Another caveat to this review concerns the fact that, while it is important to consider the baseline in γH2AX foci for proper quantification of the radio-induced DNA DSBs (and the factors are quite well reported), no mention is made to the limitations that the described techniques to detect/count foci (IF and flow cytometry) bring along.

The reviewer raises up an important aspect of these techniques. We include the limitations of the γH2AX foci count at the end of the paragraph “Methods for DSB assessment and current problems”. In particular we add these sentences:

Beside these useful applications, technical issues of γH2AX foci quantification methods have to be considered.

An important aspect is related to physical and geometrical limitations due to foci intrinsic extension, and to fluorescence microscope limit (i.e., finite resolution, multiple plane foci position and/or overlap). Moreover, energy deposition by radiation can occur in different ways, based on exposure source, determining different distribution of DSB. For example, high-LET radiation generates very close damage sites forming foci that produce a single focus signal. All these factors may determine foci signal saturation producing miscounting and making not possible to assume a 1:1 correspondence between foci count and DSB. To overcome this, computational tools to predict the real extent of the DNA damage have been developed. Considering the different parameters that determine foci saturation, these approaches quantify the miscounting of DNA DSBs after analysis of immunofluorescence images and provide a tool for a correct interpretation of DNA DSBs data.

Reviewer 2 Report

This review describes on the usage of  gammaH2AX as an evaluation tool of DNA damage. The paper reviews on the background of gammaH2AX as a DSB marker, the underlying mechanisms of H2AX phosphorylation, conditions that affect the level of H2AX phosphorylation, and methods to detect gammaH2AX etc.

The paper summarizes an interesting topic on the use of gammaH2AX, and therefore would make a significant contribution to the field and valuable for the readers in Cancers. However, this referee would like to address several points before the paper is considered to be published in Cancers.

Major points

1.       Some of the contents are somewhat difficult to follow, as the structure of the text seems not the best for the readers. Putting Introduction and the underlying phosphorylation mechanisms in the first and second chapter is OK, but the relationship between chapters of “Different conditions responsible…” and the latter chapters may not be so clear.

To get around this, authors may restructure the text and reorganize the chapters, for instance

1.     Introduction

2.     Basic mechanism of H2AX phosphorylation

3.     Conditions that affect the observed level of H2AX phosphorylation

4.     Current problems/difficulties/complications in evaluating DNA damage with H2AX phosphorylation

5.     Future perspectives (What is the general view of using the gammaH2AX method? Is it satisfactory for most cases? If not, what should be done to fix the problem or to improve, or what should be considered, especially, in applying the method in clinical studies, etc.)

6.     Conclusions

Revising the organization of the chapters may perhaps help the readers to get the overall picture of the current state of gammaH2AX method in evaluating DNA damage in cells (and a better fit with the title of the article as well).

2.       In Chapter 3. 3.1.1, “H2AX phosphorylation in the absence of DSBs” with reference to dim and bright foci. Although very important, this topic doesn’t seem to sit well with this chapter, as the content is not necessarily “a different condition” as mentioned in the title of the chapter. It may be better if the authors could consider moving this topic to another chapter, preferably to Chapter 2 (which states the mechanism of H2AX phosphorylation) for example.

Minor points

1.       Chapter 3, 3.1.1, page 6, line 160

Authors defines the brightness of foci by stating the foci has at least 35 gray levels in 8-bit scale. This seems a bit subjective as gray levels would change by other factors, such as exposure time etc?

2.       Chapter 3, 3.1.1, page 7, line 164, 166, 170

Could the authors elaborate on “replication site”? Is it “a site that the replication fork has stalled”?

3.       Chapter 3, 3.1.2, page 7, line199

What is “a rough H2AX nuclear staining”?

Author Response

Major points

1)         Some of the contents are somewhat difficult to follow, as the structure of the text seems not the best for the readers. Putting Introduction and the underlying phosphorylation mechanisms in the first and second chapter is OK, but the relationship between chapters of “Different conditions responsible…” and the latter chapters may not be so clear.

We followed the reviewer’s indication to better restructure the manuscript and now the paragraphs are:

1.Introduction

2.Basic mechanism of H2AX phosphorylation

3.Conditions that affect the observed level of H2AX phosphorylation

4.Methods for DSB assessment and current problems

  1. Applications of γH2AX in environmental risk detection and in clinical procedures
  2. Conclusions

  1. Future perspectives (What is the general view of using the gamma H2AX method? Is it satisfactory for most cases? If not, what should be done to fix the problem or to improve, or what should be considered, especially, in applying the method in clinical studies, etc.).

We have discussed these issues in “Methods for DSB assessment and current problems” (we have described the use and the different problems/advantages of manual and automatic detection of γH2AX foci) but, following referee’s suggestion, we have recalled these questions in the conclusions. In particular we add this sentence: “Currently the immunofluorescence and the use of γH2AX as a biomarker allow to evaluate the DNA DSB damage in different situations. The need for an evaluation in the clinical/diagnostic field where the IR used doses are low is satisfied by the sensitivity of the method but, in this case, unspecific signal due to the particular conditions described in this review is more conditioning and may drive to an overestimation of DNA DSB damage. Thus, when low-dose damage is evaluated, it is particularly important to discriminate unspecific signal from foci pattern due to IR exposure. In this case the manual method is more suitable as it allows a critical evaluation, assuming that the operator knows how to distinguish between unspecific signal and induced foci”.

2)           In Chapter 3. 3.1.1, “H2AX phosphorylation in the absence of DSBs” with reference to dim and bright foci. Although very important, this topic doesn’t seem to sit well with this chapter, as the content is not necessarily “a different condition” as mentioned in the title of the chapter. It may be better if the authors could consider moving this topic to another chapter, preferably to Chapter 2 (which states the mechanism of H2AX phosphorylation) for example.      

The reviewer is right. We have moved this topic to chapter “H2AX phosphorylation and the cell cycle” where Rybak et al (62) describe the nature of dim and bright foci and analyze their correlation with DNA replication sites.

Minor points

1)         Authors defines the brightness of foci by stating the foci has at least 35gray levels in 8-bit scale. This seems a bit subjective as gray levels would change by other factors, such as exposure time etc?

The indicated brightness and size parameters refer specifically to the study described in the paragraph. As correctly observed by the reviewer, they depend on the method used by the authors of the study, and they are not an absolute parameter. Rybak et al. want to shed light on the presence of two different foci populations, characterized by two distinct nature, rather than provide absolute parameter.

From the reviewer’s comment we realized that the original text was misleading and we decided to not include the parameters. We described that the authors individuated two distinct populations of foci with different size and brightness and different biological nature. 

2)           Could the authors elaborate on “replication site”? Is it “a site that the replication fork has stalled”?

We have included the definition in the text. Replication sites, DNA sites on which the replication machinery is active and 5-Ethynyl-2’-deoxyuridine (Edu) is incorporated.

3)            Chapter 3, 3.1.2, page 7, line199

What is “a rough H2AX nuclear staining”?

It means an irregular staining. As described in the study, during S-phase γH2AX staining is irregular and does not produce distinct foci signal. We decided to use the word adopted by Hernández et al, but, since from the reviewer’s comment we realized that “rough” might be a misleading word, we decided to change it in “irregular”.   

Round 2

Reviewer 2 Report

The authors have adequately revised their manuscript and there will be no further comments from this referee.